# Correlation between Quality of Life under Treatment and Current Life Satisfaction among Cancer Survivors Aged 75 Years and Older Receiving Outpatient Chemotherapy in Ishikawa Prefecture, Japan

**DOI:** 10.3390/healthcare10101863

**Published:** 2022-09-24

**Authors:** Yoshiko Kitamura, Hisao Nakai, Tomoe Hashimoto, Yuko Morikawa, Yoshiharu Motoo

**Affiliations:** 1School of Nursing, Kanazawa Medical University, 1-1 Uchinada, Kahoku 920-0265, Ishikawa, Japan; 2Komatsu Sophia Hospital, Komatsu 923-0861, Ishikawa, Japan

**Keywords:** cancer survivors, chemotherapy, personal satisfaction, quality of life

## Abstract

Life satisfaction is increasingly important for older cancer survivors as the global population ages and the life expectancy 29 of cancer survivors increases. This study sought to identify factors associated with physical symptoms, quality of life under treatment, and current life satisfaction in cancer survivors aged 75 years and older receiving outpatient chemotherapy. Information about treatment for cancer survivors was collected from electronic medical records, and interviews were conducted to assess life satisfaction under treatment. Participants were older cancer survivors in Ishikawa, Japan. Of the participants, 80% lived on the Noto Peninsula. The average linear distance traveled for treatment was 40.7 km. The factors associated with patients’ dissatisfaction with their current lives included general malaise (odds ratio: 9.61; 95% confidence interval: 1.28–72.22) and being less happy now than when they were younger (odds ratio: 10.559; 95% confidence interval: 1.50–74.24). In outpatient cancer treatment for survivors aged 75 years and older, support should consider the distance to the hospital. As in past studies, general malaise was shown to have a negative impact on the lives of cancer survivors aged 75 years or older. Support providers should pay attention to patients’ general malaise when providing support.

## 1. Introduction

The number of cancer survivors continues to increase because of advances in early detection and treatment, in addition to the aging and growth of the population [1]. Estimates up to 2035 predict an increase in the number of older cancer patients worldwide [2]. In the United States, the number of older cancer survivors continues to increase, and effective interventions to address the complex needs of older cancer survivors are currently lacking [3]. Older cancer survivors often live with existing or developing comorbidities and have complex medical and psychosocial care needs [4,5].

The maintenance and improvement of physical function and health-related quality of life (QOL) in the treatment of older cancer survivors are increasingly regarded as hard endpoints for clinical cancer research [6]. Regarding QOL and happiness in older cancer survivors, a study of people aged 75 years and older in Sweden reported that people with cancer had lower QOL than people without cancer [7]. Older breast cancer survivors were found to have lower QOL than younger women, and comorbidities and socioeconomic status have also been reported to affect QOL [8]. Moreover, older breast cancer survivors are reported to have lower life satisfaction, mastery, and spiritual well-being than younger survivors [9]. Many studies of satisfaction among older cancer survivors have examined satisfaction with care and medical treatment. Cancer survivors who recognize that they are being treated with courtesy and respect by their providers [10] and are receiving adequate information and communication from providers have been shown to report greater satisfaction with health care [11,12,13].

Reports of QOL under treatment and life satisfaction of older cancer survivors [14] revealed that, among survivors aged 60 years and older, social support from friends and family is a predictive factor for social well-being [15]. Furthermore, health-related QOL for cancer survivors and their spouses or partners has been shown to affect satisfaction with cancer treatment outcomes [16]. Although population projections of global cancer patients by age group indicate an increase in the number of cancer patients aged 75 years and older [2], only a small number of studies have examined cancer survivors in this age group. For example, one study examined mental health in breast cancer survivors aged 65 years and older [17]. Approximately half of the participants in the study were aged 75 years and older, but they were not analyzed by age group [17]. In eastern Europe, relative survival rates for breast cancer were reported to decrease rapidly with age (75 years and older) [18]. Risk factors for psychosocial well-being and psychosocial problems in cancer survivors aged 65 years and older have been examined, but no analyses focusing on those aged 75 years and older have been reported [19]. Importantly, no previous studies have examined QOL under treatment or post-treatment symptoms among older cancer survivors aged 75 years and older while also examining factors related to life satisfaction.

Japan’s population is aging at an unprecedented rate. Considering that the number of older people is predicted to continue to increase in Japan until 2036 [20] and that the life expectancy of cancer patients is improving [21], it is important to understand the symptoms that cancer survivors aged 75 years and older are living with, and the factors that affect their life satisfaction, to identify the most effective support to provide to them.

The purpose of the current study was to evaluate the correlation between physical symptoms and QOL under treatment of cancer patients aged 75 years and older receiving outpatient chemotherapy in Ishikawa Prefecture, Japan, and their current life satisfaction. By studying the physical symptoms and QOL under treatment of cancer survivors aged 75 years and older, and by identifying factors related to satisfaction with their current lives, it was possible to obtain survivors’ suggestions for improving their QOL for the remainder of their lives. This report from Japan, in which population aging is progressing at a more rapid pace compared with that in the rest of the world, will serve as a reference for other countries in which population aging is receiving increasing attention.

## 2. Methods

### 2.1. Ethical Considerations

This research was conducted in accordance with the Declaration of Helsinki, 1995 (as revised in Seoul, 2008), and was carried out with the consent of the University Medical Research Ethics Review Committees at the authors’ universities (No. I691). Prior to the interviews, we explained to participants, verbally and in writing, the purpose and significance of the study, the research method, that participation was voluntary, and that individuals would not be identified when the results are published.

### 2.2. Data Collection

We studied patients aged 75 years and older who were receiving outpatient chemotherapy at Kanazawa Medical University Hospital in Ishikawa Prefecture, Japan. Ishikawa Prefecture is located in the center of Hokuriku facing the Sea of Japan, with the Noto Peninsula in the north protruding into the Sea of Japan [22]. The population of Ishikawa Prefecture is approximately 1.12 million [23], and approximately 30% of the population is aged 65 years and older [24].

This study utilized a cross-sectional survey method. Data were collected through hospital medical records and interviews with participants. Interviews were conducted by two people: a research representative and a research collaborator. The research collaborator underwent interview training. We created an original questionnaire for this research. To explore the physical symptoms of older cancer survivors, we created questionnaire items by referencing common cancer and treatment-related symptoms listed by the National Cancer Center Japan [25]. Items from the Barthel Index [26,27] were used to examine the current activities of daily living. The Philadelphia Geriatric Center Morale Scale [28] and positive or negative feelings of morale [29] items were used to interview patients regarding overall satisfaction and current life satisfaction. The survey was conducted from 1 March to 6 April 2022.

### 2.3. Survey Details

Table 1 shows the survey items.

### 2.4. Analysis Methods

We analyzed the responses of participants who answered all of the following items: annual income, cancer insurance coverage, support providers, degree of current physical symptoms, activities of daily living, self-care behavior, and life satisfaction.

To understand participants’ characteristics, the median and range deviations were calculated for age and time since their cancer diagnosis. We calculated the distribution of treatment history by frequency of hospital visits, underlying diseases, and treatment methods. We created a spider graph that connected a straight line from Kanazawa Medical University Hospital to the government office of the participants’ residences and calculated the geographical distribution. ESRI ArcGIS Pro (ESRI; Redlands, CA, USA) was used to analyze geographic distribution.

To evaluate the factors affecting current life satisfaction, for age, the median age was used to categorize participants into two categories of below and above the age of 80. The median time since cancer diagnosis was used to categorize participants into two categories: under and over 17.5 months. Regarding annual income, 2.8 million yen (the base amount at which the self-pay ratio for long-term care insurance services in Japan goes from 10% to 20%) was used, with participants classified into the two categories of “2.8 million yen or above” and “less than 2.8 million yen or unknown”, with the latter classified as “Other”.

For cancer insurance coverage, “I have insurance” was classified as “Yes”, and “I don’t have insurance/I don’t know” were classified as “Other”. For the degree of physical symptoms, “many symptoms/symptoms present” was classified as “Yes” and “a few symptoms/no symptoms” were classified as “Other”. For treatment response, “independent” was classified as “Independent” and “being watched over/partial assistance/total assistance” was classified as “Other”. For life satisfaction, “strongly agree/agree” was classified as “Agree” and “disagree/strongly disagree” was classified as “Disagree”.

The correlation between satisfaction with current life and basic attributes, annual income, cancer insurance coverage, support providers, time since cancer diagnosis, metastasis, underlying disease, medication status, degree of physical symptoms, self-care behavior, and life satisfaction were assessed using the Chi-square test or Fisher’s exact test.

With satisfaction with current life as the dependent variable, a binary logistic regression analysis was performed with a forced entry of sex and cancer insurance coverage as covariates, and general malaise and happier now than when young (variables with *p* < 0.05 in univariate analysis) and unable to sleep (variables with *p* < 0.06 in univariate analysis). The significance level was set at 5%. SPSS version 27 (IBM Corporation; Armonk, NY, USA) was used for all statistical analyses.

## 3. Results

A total of 62 people were surveyed. Of these, 50 patients (80.6%) were included in the analysis, excluding seven patients who discontinued treatment on the day, three patients who did not agree to participate in the survey, and two patients who discontinued their responses.

Participants’ median age (range) was 78.5 (75–86) years old. The median time since cancer diagnosis (range) was 17.5 (2–240) months. Activities of daily living responses were as follows: one respondent was independent for excretion and straightening their posture, and required being watched over for dressing; two respondents required being watched over and two required partial assistance for bathing; one participant required being watched over for eating meals; eight respondents required being watched over, six required partial assistance, and one required total assistance for going up and down the stairs; and seventeen respondents required partial assistance and thirteen required total assistance for meal preparation. Table 2 shows the treatment history by the frequency of visits, underlying diseases, and treatment method. Table 3 shows the different combinations of treatment modalities. Figure 1 shows the distribution of cancer sites.

Of the 50 participants, 40 lived in the Noto Peninsula. Of these, ten participants lived in Kahoku, seven lived in Hakui, and six lived in Tsubata (Figure 2). The average Euclidean distance from Kanazawa Medical University Hospital to the government office in the participants’ area of residence was 40.7 km, with the furthest municipality being Suzu at 100.6 km. A spider graph is shown in Figure 2.

A total of 29 (58.0%) respondents were aged between 75 and 80 years, and 21 (42.0%) were aged 80 years or older. There were 35 men (70.0%) and 15 women (30.0%).

The results of the univariate analysis are shown in Table 4. The proportion of respondents who were dissatisfied with their present life was significantly higher in seven respondents (70.0%) with general malaise (*p* = 0.001), ten respondents (47.6%) who did not think they were happier now than when they were younger (*p* = 0.003), and eight respondents (42.1%) who could not sleep (*p* = 0.054).

Table 5 shows the results of binary logistic regression analysis with dissatisfaction with current life as the dependent variable. In controlling for the effects of sex and cancer insurance coverage, the related factor for dissatisfaction with current life was “Yes” as opposed to “Other” for general malaise (odds ratio: 9.61; 95% confidence interval: 1.28–72.22), and “Agree” rather than “Disagree” for feeling happier now than when they were younger (odds ratio: 10.56; 95% confidence interval: 1.50–74.24) (Table 5).

## 4. Discussion

A survey of cancer patients in Japan in 2019 reported that the large intestine was the most common cancer site, followed by the lungs, stomach, and breasts [31] (Japan Cancer Information Service). According to the results for the Ishikawa Prefecture (the target area of this study) in 2016, the stomach was the most common site of cancer among registrants aged 75 and older, excluding epithelial cancer, followed by the large intestine, lungs, and colon. Issues with the digestive system were the most common symptoms in relation to cancer. Considering that the colon was also the most common site of cancer in this study, the trend seems to be similar to that among patients aged 75 years and older in Ishikawa. However, because the report indicated that cancer in the stomach, lungs, and colon (in descending order) were most common among men, and cancer in the large intestine, colon, and stomach (in descending order) were most common among women [30], lung cancer may have been ranked as second-most common in the current study because 70% of the participants were male. Kanazawa Medical University Hospital is located at the entrance of the Noto Peninsula and has been designated as a regional cancer medical care base hospital, with medical care partnerships with medical institutions on the Noto Peninsula [32] and a specialized outpatient clinic for hematology. These characteristics may have contributed to the finding that hematological cancer was the third most common cancer type in the current study. The fact that 80% of participants were patients from the Noto Peninsula may therefore be attributed to the location of the Kanazawa Medical University Hospital.

For patients receiving outpatient chemotherapy, transportation costs for those living far from treatment sites have become a major burden [33,34]. In addition, in Japan, the physical burden and fatigue of patients who must travel a long distance to visit hospitals are significant issues [35]. A study conducted in Iowa reported that patients living in areas without a chemotherapy provider had an average driving time of 58 min to receive treatment, compared with 21 min for patients living near a dedicated chemotherapy facility [36]. A study in North Carolina indicated that people who lived in rural areas more than 32 km from their chemotherapy provider were less likely to receive chemotherapy [37]. Travel distance is an important factor in outpatient cancer care in Ishikawa Prefecture, given that the study population comprised cancer survivors 75 years of age and older, the average distance of hospital visits was 40 km or more, and 80% of participants were Noto Peninsula residents. In Japan, serious accidents caused by driving errors by people aged 75 years and older have become a problem [38]. It is important to reduce the burden of hospital visits for older cancer survivors to receive chemotherapy when considering the combined symptoms of cancer, complications arising from treatment, and chemotherapy sequelae among older cancer survivors.

In Japan, 75-year-old cancer survivors have a median life expectancy of 11.9 years for men and 16.1 years for women [39]. It is important that older cancer survivors spend their remaining time satisfactorily while coping with their disease. Many cancer survivors suffer from aftereffects of cancer treatment [40]. After primary cancer treatment is completed, some symptoms continue to have a negative impact on cancer survivors for years to come. Such symptoms include physical limitations, cognitive limitations, depression, anxiety, sleep problems, fatigue, pain, and sexual dysfunction [41,42,43]. In particular, fatigue can persist for a long time and seriously impact QOL [44]. As previously reported, fatigue may affect life satisfaction in older cancer survivors aged 75 years and older. The response to the question of whether participants are happier now than when they were younger is a self-reported answer that involves recollection of their past condition before they had cancer. The effects of the respondents’ life history and experiences (e.g., war) cannot be controlled for in comparisons between when they were younger and the present day; thus, from the current results, we cannot extrapolate a correlation between respondents thinking that they are less happy now than when they were younger and not being satisfied with their present lives. However, service providers serving older cancer survivors aged 75 years and older may be able to help survivors feel more satisfied with the rest of their lives by addressing the issues that lead them to complain that they are less happy now than when they were younger.

The current study involved several limitations. First, because we included only 50 people from one hospital, the generalizability of the study results may be limited. Second, because the current study used a self-reported survey that was conducted at an outpatient clinic, there may have been differences in the answers depending on the participant’s knowledge, cognitive function, and physical condition on the day. Third, in meal preparation among activities of daily living, it was not possible to distinguish between the simple evaluation of activities and cases in which the participant does not regularly prepare meals by themselves because their spouse typically takes care of meal preparation. Finally, because this study used a cross-sectional study design, it is not possible to establish causal relationships between the variables under investigation.

## 5. Conclusions

In outpatient chemotherapy for older cancer survivors, it is recommended that support be provided by considering the travel distance required for treatment. As in previous studies, fatigue has been shown to impact the lives of cancer survivors aged 75 years and older. Thus, it is recommended that support providers consistently focus on the presence of fatigue while providing support.

These results suggest that some older cancer survivors with fatigue may spend the rest of their lives feeling unsatisfied with their situation. For older cancer survivors to be satisfied with their current lives, it is necessary to clarify the characteristics of fatigue they exhibit and to develop appropriate care. At present, it is recommended that support providers consistently focus on the presence of fatigue while providing support and careful observation.

## Figures and Tables

**Figure 1 healthcare-10-01863-f001:**
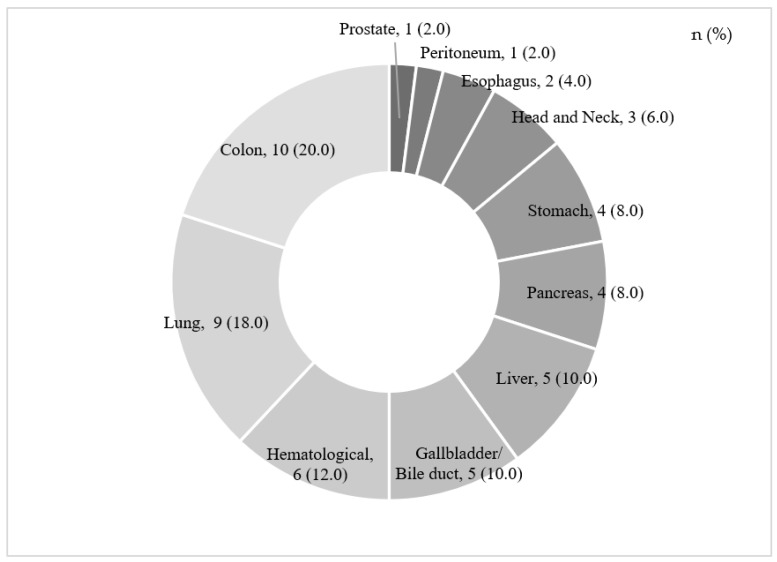
Cancer site (*n* = 50).

**Figure 2 healthcare-10-01863-f002:**
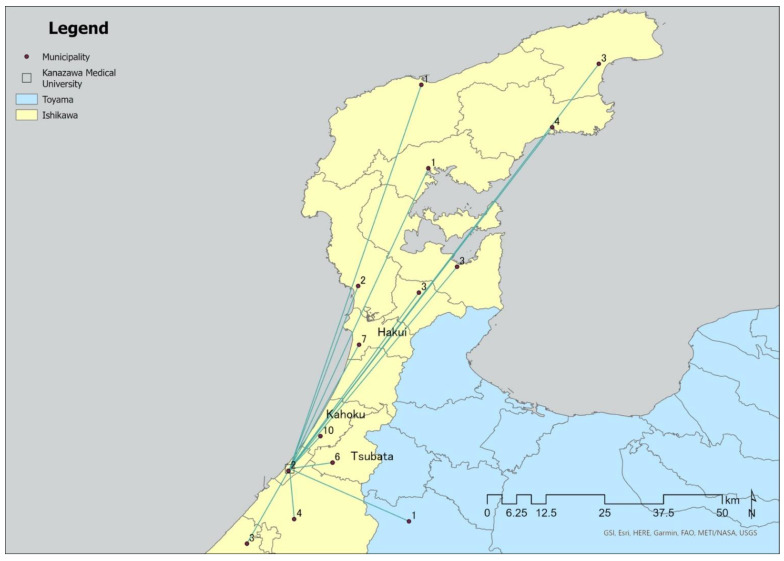
Spider graph.

**Table 1 healthcare-10-01863-t001:** Survey items.

Items obtained from hospital medical records		
	Background	
		Information included age, sex, municipality of residence, cancer site, time since cancer diagnosis, and presence or absence of metastases.
	The frequency of visits	
		Response options were “every week”, “every 2 weeks”, “every 3 weeks”, or “other”.
	Underlying diseases	
		Response options were diseases of the “circulatory system”, “respiratory system”, “cranial nerve system”, “endocrine system”, “renal/urinary system”, “motor system”, “sensory system”, “other”, and “none” (multiple answers allowed).
	Past treatment according to treatment method	
		Response options were “surgery”, “radiation therapy”, “chemotherapy”, and “other” (multiple answers allowed).
	Medication status	
		This item was responded to as a “yes/no” question for “oral medications”, “opioids”, “non-opioids”, “cancer therapeutic drugs”, “sleeping pills”, “hypertension medication”, “diabetes drugs”, and “laxatives”.
Items for which participants were asked to report details		
	Annual income	
		Responses were classified into three categories: “less than 2.8 million yen”, “between 2.8 million and 3.4 million yen”, “3.4 million yen or more”, and “I don’t know” [30], referring to the self-pay ratio of long-term care insurance service charges in Japan.
	Cancer insurance coverage	
		Responses were classified into three categories: “I have insurance”, “I don’t have insurance”, and “I don’t know”.
	Support providers	
		Participants responded in a “yes/no” format regarding “cohabitants (people who lived with them)”, “relatives nearby”, and “people who would rush to help in an emergency”.
	Physical symptoms	
		Responses included “general malaise”, “loss of appetite”, “pain”, “constipation”, “respiratory distress”, and “other”, which respondents reported using four options: “strongly present”, “present”, “slightly present”, and “not present”.
	Activities of daily living	
		Responses included “I can go up and down the stairs”, “prepare meals”, “eat meals by myself”, “bathe by myself”, “urinate by myself”, “defecate by myself”, “change clothing by myself”, and “straighten my posture by myself”, and were classified into four categories: “independent”, “need to be watched over”, “partial assistance”, and “total assistance”.
	Self-care behaviors	
		Responses included “being able to see a doctor at a designated date and time”, “being able to go to the hospital by oneself”, and “taking oral medicines as instructed”, which were classified into four categories: “independent”, “need to be watched over”, “partial assistance”, and “total assistance”.
	Satisfaction with life	
		“I started to worry about small things”, “I think I’m happier now than when I was young”, “I’m unable to sleep”, “I get agitated easily”, “I feel lonely”, and “I feel uncomfortable with my family” were reported using four response options: “strongly agree”, “agree”, “disagree”, and “strongly disagree”.
	Satisfaction with current life	
		Responses were classified using four response options: “strongly agree”, “agree”, “disagree”, and “strongly disagree.

**Table 2 healthcare-10-01863-t002:** Participants’ characteristics (*n* = 50).

Item		*n*	(%)
Age (median (range))	78.5 (75–86)		
Time since cancer diagnosis (median (range))	17.5 (2–240)		
Frequency of visits			
	Every week	3	(6)
	Every 2 weeks	19	(38)
	Every 3 weeks	18	(36)
	Other	10	(20)
Underlying diseases (multiple answers allowed)			
	Circulatory system	30	(60)
	Endocrine system	23	(46)
	Urinary system	8	(16)
	Motor system	7	(14)
	Sensory system	6	(12)
	Respiratory system	6	(12)
	Others	12	(24)
Treatment history by treatment method (multiple answers allowed)			
	Chemotherapy	49	(98)
	Surgery	27	(54)
	Radiation therapy	13	(26)

**Table 3 healthcare-10-01863-t003:** Percentage of combinations of treatment modalities (multiple answers allowed) (*n* = 50).

Treatment Method	*n*	(%)
Surgery + Chemotherapy	20	(40.0)
Chemotherapy	17	(34.0)
Surgery + Chemotherapy + Radiation therapy	7	(14.0)
Chemotherapy + Radiation therapy	6	(12.0)

**Table 4 healthcare-10-01863-t004:** Association between the quality of life under treatment and current life satisfaction of cancer survivors.

				Satisfaction with Current Life				
Item	Category	Total		Agree		Disagree		
		N	%	N	%	N	%	*p*-Value
Basic attributes, annual income, cancer insurance coverage, support provider								
Age	<80	29	58.0	22	75.9	7	24.1	0.724 ^a^
	≥80	21	42.0	15	71.4	6	28.6	
Sex	Male	35	70.0	24	68.6	11	31.4	0.294 ^b^
	Female	15	30.0	13	86.7	2	13.3	
Annual income	2.8 million yen or above	16	32.0	11	68.8	5	31.3	0.731 ^b^
	Other	34	68.0	26	76.5	8	23.5	
Cancer insurance coverage	Yes	28	56.0	21	75.0	7	25.0	0.856 ^a^
	Other	22	44.0	16	72.7	6	27.3	
Co-habitants	Yes	43	86.0	33	76.7	10	23.3	0.357 ^b^
	No	7	14.0	4	57.1	3	42.9	
Relatives nearby	Yes	7	14.0	4	57.1	3	42.9	0.357 ^b^
	No	43	86.0	33	76.7	10	23.3	
People who would rush to help in an	Yes	49	98.0	36	73.5	13	26.5	1.000 ^b^
emergency	No	1	2.0	1	100.0	0	0.0	
Time since cancer diagnosis, metastasis, underlying disease, medication status								
Time since cancer diagnosis	< 17.5	25	50.0	19	76.0	6	24.0	0.747 ^a^
	≥17.5	25	50.0	18	72.0	7	28.0	
Metastasis, cancer in other sites	Yes	24	48.0	19	79.2	5	20.8	0.424 ^a^
	No	26	52.0	18	69.2	8	30.8	
Underlying disease	Yes	46	92.0	35	76.1	11	23.9	0.275 ^b^
	No	4	8.0	2	50.0	2	50.0	
Oral medication	Yes	49	98.0	36	73.5	13	26.5	1.000 ^b^
	No	1	2.0	1	100.0	0	0.0	
Opioids	Yes	1	2.0	0	0.0	1	100.0	0.260 ^b^
	No	49	98.0	37	75.5	12	24.5	
Non-opioids	Yes	15	30.0	12	80.0	3	20.0	0.728 ^b^
	No	35	70.0	25	71.4	10	28.6	
Cancer therapeutic drugs	Yes	13	26.0	11	84.6	2	15.4	0.469 ^b^
	No	37	74.0	26	70.3	11	29.7	
Sleeping pills	Yes	6	12.0	5	83.3	1	16.7	1.000 ^b^
	No	44	88.0	32	72.7	12	27.3	
Hypertension medication	Yes	21	42.0	14	66.7	7	33.3	0.314 ^a^
	No	29	58.0	23	79.3	6	20.7	
Diabetes drugs	Yes	9	18.0	6	66.7	3	33.3	0.679 ^b^
	No	41	82.0	31	75.6	10	24.4	
Laxatives	Yes	27	54.0	20	74.1	7	25.9	0.990 ^a^
	No	23	46.0	17	73.9	6	26.1	
Physical symptoms								
General malaise	Present	10	20.0	3	30.0	7	70.0	0.001 ^b^
	Other	40	80.0	34	85.0	6	15.0	
Loss of appetite	Present	15	30.0	10	66.7	5	33.3	0.493 ^b^
	Other	35	70.0	27	77.1	8	22.9	
Pain	Present	14	28.0	11	78.6	3	21.4	0.734 ^b^
	Other	36	72.0	26	72.2	10	27.8	
Constipation	Present	17	34.0	10	58.8	7	41.2	0.099 ^b^
	Other	33	66.0	27	81.8	6	18.2	
Respiratory distress	Present	6	12.0	4	66.7	2	33.3	0.643 ^b^
	Other	44	88.0	33	75.0	11	25.0	
Other	Present	19	38.0	15	78.9	4	21.1	0.742 ^b^
	Other	31	62.0	22	71.0	9	29.0	
Self-care treatment								
Being able to see a doctor at a	Independent	45	90.0	33	73.3	12	26.7	1.000 ^b^
designated date and time	Other	5	10.0	4	80.0	1	20.0	
Being able to go to the hospital by	Independent	25	50.0	17	68.0	8	32.0	0.333 ^a^
oneself	Other	25	50.0	20	80.0	5	20.0	
Taking oral medicines as instructed	Independent	44	88.0	32	72.7	12	27.3	1.000 ^b^
	Other	6	12.0	5	83.3	1	16.7	
Satisfaction in life								
I started to worry about small things.	Agree	15	30.0	9	60.0	6	40.0	0.170 ^b^
	Disagree	35	70.0	28	80.0	7	20.0	
I think I’m happier now than when I	Agree	29	58.0	26	89.7	3	10.3	0.003 ^a^
was young.	Disagree	21	42.0	11	52.4	10	47.6	
I’m unable to sleep.	Agree	19	38.0	11	57.9	8	42.1	0.054 ^b^
	Disagree	31	62.0	26	83.9	5	16.1	
I get agitated easily.	Agree	5	10.0	4	80.0	1	20.0	1.000 ^b^
	Disagree	45	90.0	33	73.3	12	26.7	
I feel lonely.	Agree	10	20.0	5	50.0	5	50.0	0.101 ^b^
	Disagree	40	80.0	32	80.0	8	20.0	
I feel uncomfortable with my family.	Agree	14	28.0	8	57.1	6	42.9	0.149 ^b^
	Disagree	36	72.0	29	80.6	7	19.4	

Aged 75 Years and older receiving outpatient chemotherapy. ^a^ χ^2^ test, ^b^ Fisher’s exact test.

**Table 5 healthcare-10-01863-t005:** The impact of general malaise, “I think I’m happier now than when I was young”, and “I’m unable to sleep” on current life satisfaction.

Item	Category	OR	95% CI		*p*-Value
			Lower Limit	Upper Limit	
Sex	Male/Female	2.35	0.26	21.22	0.447
Cancer insurance coverage	Yes/ Other	0.92	0.11	7.44	0.938
General malaise	Yes/Others	9.61	1.28	72.22	0.028
I think I’m happier now than when I was young.	Disagree/Agree	10.56	1.50	74.24	0.018
I’m unable to sleep.	Agree/Disagree	3.16	0.53	18.92	0.208

## Data Availability

The data analyzed during this study are included in this published article. Further inquiries can be directed to the corresponding authors.

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
