# Peer review of "Correlation between Quality of Life under Treatment and Current Life Satisfaction among Cancer Survivors Aged 75 Years and Older Receiving Outpatient Chemotherapy in Ishikawa Prefecture, Japan"

_healthcare, 2022, doi:10.3390/healthcare10101863_

Round 1

Reviewer 1 Report

Dear Authors,

the manuscript is very interesting. However, I have some concerns.

1) The abstract should be rewritten. Introduction is not clear enough. Why life satisfaction is that important for cancer survivors? 

2) Change the keywords. Use only MeSH terms.

3)The introduction is too long. It includes a lot of irrelevant information. Please, shorten it and limit only to necessary information.

4) Figure 1 is not needed. I think it should be removed.

5) Lines 109-113 can be omitted.

6) Information from paragraph 2.2 should be presented in table.

7) Pragaraph 2.4 should be at the beginning of the Materials and Methods

8) I would suggest to present type of received treatment in separate table. The table should also include different combinations of treatment modalities.

9) Conclusions should be limited only to the most important issues from your manuscript. Please modify.

10) Line 336 - what does "please add:" stand for?

11) I suggest adding the reference: 

Mitus-Kenig M, Derwich M, Czochrowska E, Pawlowska E. Quality of Life in Orthodontic Cancer Survivor Patients-A Prospective Case-Control Study. Int J Environ Res Public Health. 2020 Aug 12;17(16):5824. doi: 10.3390/ijerph17165824.

Reviewer 2 Report

interesting article that mets the scope of the journal

creal structure, sound methodology, appropriate writting tone. no syntax and grammar errors.

approprate presentation of references.

Argumentation supported by bibliography。

which are the factors associated with physical symptoms, quality of life under treatment, and current life satisfaction can be identified in cancer survivors aged 75 years and older receiving outpatient 14 chemotherapy.

it is original and relevant and can be associated with the newest trends of lifestyle medicine and integrative oncology 

there are only very few relevant studies so it offers new ground  of knowledge. However, the sample could be ideally bigger, they have 65 patients. And the conclusions are consistent with the evidence and arguments presented and they address the main question posed. The references are appropriate

The tables and figures are in an appropriate format.

Reviewer 3 Report

Dear editor and dear authors,

Thank you for the opportunity to review your paper entitled “Correlations between Quality of life under treatment and current life satisfaction among cancer survivors aged 75 years and older receiving outpatient chemotherapy in Ishikawa Prefecture, Japan.”

I appreciated the paper; here are some suggestions:

The introduction, methodology, results, and discussion sections are well framed. In the results section, page 6, lines 225-226, please check the geographic distribution of the participants. The sample is 50, not 62.

Limitations are identified. We suggested improving the conclusions. I recommend that the authors explain to readers the significant impact of these results in clinical practice and which suggestions could be given to the patients for improving their QOL.

Thank You
